# Ischemic Etiology and Prognosis in Men and Women with Acute Heart Failure

**DOI:** 10.3390/jcm10081713

**Published:** 2021-04-15

**Authors:** Lourdes Vicent, Jose Guerra, Rafael Vazquez-García, José R. Gonzalez-Juanatey, Luis Martínez Dolz, Javier Segovia, Domingo Pascual-Figal, Ramón Bover, Fernando Worner, Juan Delgado, Francisco Fernández-Avilés, Manuel Martínez-Sellés

**Affiliations:** 1Cardiology Department, Hospital Universitario 12 de Octubre, 28028 Madrid, Spain; mlourdesvicent@gmail.com (L.V.); juan.delgado@salud.madrid.org (J.D.); 2Cardiology Department, Hospital de la Santa Creu i Sant Pau, 08041 Barcelona, Spain; JGuerra@santpau.cat; 3CIBER de Enfermedades Cardiovasculares (CIBERCV), Instituto de Salud Carlos III, 28029 Madrid, Spain; Jose.Ramon.Gonzalez.Juanatey@sergas.es (J.R.G.-J.); jsecu@telefonica.net (J.S.); faviles@secardiologia.es (F.F.-A.); 4Cardiology Department, Puerta del Mar University Hospital, 11009 Cádiz, Spain; rafael.vazquez.sspa@juntadeandalucia.es; 5Cardiology Department, University Hospital, 15076 Santiago de Compostela, Spain; 6Cardiology Department, University Hospital La Fe (CIBERCV), 46026 Valencia, Spain; luismartinezdolz@gmail.com; 7Cardiology Department, Hospital Universitario Puerta de Hierro Majadahonda, 28222 Madrid, Spain; 8Cardiology Department, Hospital Virgen de la Arrixaca, Department of Medicine, University of Murcia, 30120 Murcia, Spain; dpascual@um.es; 9Cardiology Department, Hospital Clínico San Carlos, 28040 Madrid, Spain; ramonboverfreire@gmail.com; 10Servicio de Cardiología, Hospital Universitari Arnau de Vilanova, Institut de Recerca Biomèdica de Lleida, 25198 Lleida, Spain; fworner.lleida.ics@gencat.cat; 11Cardiology Department, Instituto de Investigación, Hospital General Universitario Gregorio Marañón, 28007 Madrid, Spain; 12Facultad de Medicina, Universidad Complutense, 28040 Madrid, Spain; 13Facultad de Medicina, Universidad Europea, 28670 Madrid, Spain

**Keywords:** heart failure, mortality, ischemic heart disease

## Abstract

Coronary heart disease is common in heart failure (HF). Our aim was to determine the impact of ischemic etiology on prognosis among men and women with HF. This study is a prospective national multicenter registry. The primary endpoint was 12-month mortality. Patients with HF and ischemic heart disease were stratified according to sex. A total of 1830 patients were enrolled of which 756 (41.3%) were women. Ischemic etiology was more common in men (446 (41.6%)) than in women (167 (22.2%)). Among patients with ischemic HF, diabetes was more frequent in women than in men. Ischemic etiology was not associated with higher mortality risk, and this was true for women (Hazard Ratio [HR] 1.51, 95% Confidence Interval [CI] 0.98–2.32; *p* = 0.61) and men (HR 1.14, 95% CI 0.81–1.61; *p* = 0.46), *p*-value for interaction: 0.067. Mortality/readmission risk in ischemic HF increased in men with previous readmissions (HR 1.15, 95% CI 1.02–1.29; *p* = 0.022), chronic obstructive pulmonary disease (HR1.20, 95% CI 1.02–1.41; *p* = 0.026) and in women with diabetes (HR 2.23, 95% CI 1.05–4.47; *p* = 0.035). Ischemic etiology was not associated with mortality in HF patients. In ischemic HF, the variables associated with a poor prognosis were diabetes in women and previous readmissions and chronic obstructive pulmonary disease in men.

## 1. Introduction

Ischemic heart disease is the leading cause of mortality worldwide and is a major cause of premature mortality and disability [1]. Ischemic heart disease is a common condition associated with heart failure (HF) and a powerful predisposing factor [2] of developing HF [3,4]. In addition, ischemic etiology has been associated with a poor prognosis in HF patients in some studies [5], but not in others [6].

Differences related to sex have been described in HF patients, including etiology, demographic profile, medical history, and prognosis [7,8,9]. Compared to men, women have a higher prevalence of hypertension, valvular heart disease, and diabetes but a lower prevalence of ischemic heart disease [7,8,10]. However, the effect of sex on the impact of ischemic heart disease on HF prognosis is controversial [11,12]. A worse prognosis has been described in women with coronary artery disease, with an increased risk of incident HF, cardiogenic shock [12], lower frequency of revascularization, and a lower use of antiplatelet drugs, statins, and beta-blockers [10,13,14]. Moreover, women have been underrepresented in clinical trials of HF and ischemic heart disease [15].

Our objective was to assess the impact of ischemic heart disease on HF prognosis in men and women. We have also compared the characteristics and treatment of women admitted with heart failure with their male counterparts.

## 2. Methods

The present study is a sub-analysis of the Spanish Network for the Study of Heart Failure II registry (REDINSCOR II). The methodology of the study has been previously detailed [16,17]. Briefly, the REDINSCOR II is a prospective, multicenter, nationwide study including adults admitted for acute HF in 20 Cardiology departments from Spanish hospitals, from October 2013 to December 2014. According to the definition of the current clinical practice guidelines, all patients had a diagnosis of acute HF at admission [18].

Patients with HF and ischemic heart disease were stratified according to sex. We considered ischemic etiology in the presence of coronary artery disease of sufficient severity and extension to justify myocardial damage [19,20].

The primary endpoint was mortality at 12 months. We also analyzed the composite of all-cause mortality/hospital readmissions due to HF incidence of sudden cardiac death, mortality due to refractory HF, or heart transplantation at 12 months. Hospital readmissions and mortality at 1 and 6 months were also assessed.

This study was accomplished with the Declaration of Helsinki and was approved by the Ethics Committee of the recruiting hospitals (9/12/2013 CEIC: 57/2013; 19/09/2013 CEIC: 13/2013). All patients provided written informed consent.

### Statistical Analysis

Continuous variables are shown as mean (standard deviation) or median (interquartile interval) for non-normally distributed variables. Categorical data are presented as frequencies and percentages. Continuous quantitative variables were compared using Student’s *t*-test and ANOVA for the comparison of means or the Wilcoxon rank-sum in nonparametric data. Categorical variables were analyzed using the χ2 test and the Fischer exact test. Bonferroni’s correction was applied for multiple comparisons. 

Multivariate analysis included multiple logistic regression techniques and Cox regression modeling for the study endpoints. To determine which variables were entered into the final model, we used a sequential inclusion and exclusion method, with an inclusion *p* threshold lower than 0.05 and exclusion over 0.1. The final model included age, previous heart failure admissions, diabetes, glomerular filtration rate, HF therapies, rhythm, and anemia at discharge. All analyses were performed with the STATA software (StataCorp, College Station, TX, USA, version 14.0).

## 3. Results

A total of 1830 HF patients were enrolled in the registry; the mean age was 73.3 ± 10.6 years, 756 (41.3%) were women, and 613 (33.5%) had ischemic etiology. Ischemic etiology was significantly more common in men (446 [41.5%]) compared to women (167 (22.1%)), *p* < 0.001. Table 1 shows basal demographic characteristics in the studied patients according to sex and etiology. Compared to men, women were significantly older, and we found significant differences in the frequency of cardiovascular risk factors. Women with ischemic HF had a more common history of diabetes, while men more often had a history of tobacco and alcohol consumption. In patients with non-ischemic HF, the etiology of HF was valvular heart disease in 411 (33.9%), hypertension in 535 (44.1%), idiopathic/familiar in 199 (16.4%), hypertrophic cardiomyopathy in 53 (4.4%), and infiltrative diseases in 15 (1.2%).

In patients with ischemic etiology, men were carriers of an implantable cardioverter-defibrillator more frequently than women (57 (12.8%) vs. 4 (2.4%), *p* < 0.001), and this was also the case in patients with non-ischemic HF (50 (8.0%) vs. 16 (2.7%), *p* < 0.001). Concerning pharmacological treatment, women with ischemic etiology more commonly received angiotensin II receptor blockers and fewer mineralocorticoid receptor antagonists compared to men.

Left ventricular ejection fraction was higher in women with ischemic etiology, compared to men (47 ± 18% vs. 39 ± 16%) (Table 2). Men, however, had greater ventricular remodeling and dilation and a wider QRS duration. Mortality during hospital admission was comparable in men and women (43 (4.0) vs. 28 (3.7)), but length of hospital stay was shorter in women (9.8 ± 8.5 vs. 12.2 ± 17.4). Women with ischemic HF were older, had greater comorbidity, lower left ventricular ejection fraction, and more common previous HF hospitalizations compared to women with non-ischemic HF.

Unadjusted analysis of mortality and hospital readmissions showed similar outcomes during follow-up among men and women with ischemic etiology of HF (Table 3). However, women with non-ischemic etiology tended to present fewer hospital readmissions at 12 months. In adjusted Cox regression analysis, ischemic etiology was not associated with an increased mortality at 12 months in women (HR 1.52, 95% CI 0.97–2.33, *p* = 0.052) or men (HR 1.28, 95% CI 0.911–1.79, *p* = 0.155) (Figure 1). Women received a heart transplant less often compared to men in univariate comparisons (5 (0.7%) vs. 31 (2.9%), *p* = 0.05). After a multivariate analysis adjusting for age and comorbidities, we did not find sex-related differences in heart transplants at 12 months (HR 2.53, 95% CI 0.87–7.39; *p* = 0.09). The only factor associated with a lower indication of heart transplants was older age (HR 0.92, 95% CI 0.89–0.94; *p* < 0.001).

Ischemic etiology was not an independent predictor of mortality (HR 1.21, 95% CI 0.96–1.53, *p* = 0.11) (Table 4), and this was true for women (HR 1.51, 95% CI 0.98–2.32; *p* = 0.61) and men (HR 1.14, 95% CI 0.81–1.61; *p* = 0.46), interaction *p*-value: 0.067. Ischemic etiology was also not associated with the combined endpoint of mortality/readmissions. In patients with ischemic HF, diabetes was associated with an increased risk of mortality/readmissions in women and previous HF admissions and chronic obstructive pulmonary disease in men.

## 4. Discussion

We have found that, among patients admitted with HF, ischemic etiology is present in a fifth of women and about 40% of men. Ischemic etiology was not associated with an increased risk of adverse events.

### 4.1. Comorbidities and Cardiovascular Risk Factors

Women are underrepresented in clinical trials, and patients from clinical trials are highly selected. Our sample is more representative of real-world patients. Previous studies that have analyzed HF’s presentation in women have shown frequent advanced age and comorbidity [21,22,23,24]. In contrast to men, the most common form of HF in women is that with preserved left ventricular function [22]. Our results were consistent with previous experiences, as women had higher left ventricular ejection fraction and older age [8,21,25]. Regarding comorbidity and cardiovascular risk factors, women with ischemic etiology frequently had diabetes (69%), but the rate of hypertension was comparable to men (Appendix A). It is possible that high blood pressure plays a less prominent role in the development of ischemic heart disease in women, which would be more related to other cardiovascular risk factors, such as diabetes [23,26]. HF risk is higher in women with diabetes than in men with diabetes [23,26,27,28]. Analyzing the characteristics of women with ischemic etiology, we have observed that the clinical profile of these patients is more similar to men with ischemic HF than to women with HF of non-ischemic etiology, suggesting that several differences associated with sex might be, in fact, related to ischemic etiology.

### 4.2. Heart Failure Therapies

The use of a implantable cardioverter-defibrillator was higher in men than in women. This might be related, in part, to the fact that men more frequently had a left ventricular ejection fraction <35% [18,25]. However, it has also previously been stated that women with HF and a reduced left ventricular ejection fraction are less likely to be referred for cardioverter-defibrillator implantation [29]. In fact, in the specific group of patients with ischemic etiology and reduced left ventricular ejection fraction, the proportion of women with a defibrillator was also lower than men. In our study, the incidence of sudden cardiac death during follow-up was similar between men and women with ischemic HF.

Traditionally, women with HF have been under-treated compared to men. This difference has been attributed to women’s more unfavorable clinical profile, with older age and common comorbidities [30]. However, in the subgroup of women with ischemic etiology, we only found significant differences in the prescription of mineralocorticoid receptor antagonists. The lower prescription of these drugs in the cohort of our study may also be explained by the lack of indication due to the higher left ventricular ejection fraction in women [25].

### 4.3. Mortality and Hospital Readmissions during Follow-Up

We did not find significant sex differences in hospital mortality and mortality/readmissions during follow-up among patients with ischemic HF.

In some previous studies, women have shown better-adjusted survival rates than in men [16,29,31,32,33], a finding which is more marked in patients with non-ischemic HF [23,34,35]. In our registry patients, the survival advantage traditionally attributed to women with non-ischemic heart failure may have been attenuated by a more advanced disease, with more years since the diagnosis.

With regard to hospitalization for heart failure, previous experiences have described that the male sex is associated with a higher probability of readmission [36,37] and the combined endpoint of mortality/readmissions [36]. The attenuation of these sex-related differences in prognosis could be explained by HF’s etiology and the negative impact of coronary heart disease in women. Indeed, women who present with an acute myocardial infarction have an increased risk of developing heart failure during admission and follow-up, compared to men [11,12]. Mental/psychological stress was not assessed in this study. Future studies should analyze the impact of psychosocial dimension on the prognosis of patients with HF according to etiology, and sex differences.

Other variables associated with worse outcomes were comorbidities, especially diabetes mellitus or chronic obstructive pulmonary disease in women, and previous hospitalizations in men with ischemic etiology. Treatment with ACEIs/ARBs was a factor associated with a lower risk of death and readmissions, regardless of sex and the etiology of heart failure.

Women tend to receive heart transplants less often than men [38]. In the univariate analysis of our study, heart transplants were performed less frequently in women. However, after multivariate adjustment, there were no differences between men and women with ischemic or non-ischemic etiology of HF. Advanced age was the only factor associated with a lower probability of receiving a heart transplant, and therefore, after statistical adjustment, the indication for transplant was similar in both sexes.

Our study has some limitations. The follow-up phase was 12 months, and a longer follow-up period may have shown significant differences in outcomes according to HF etiology and sex. The dosage of disease-modifying drugs was not recorded, and we do not have information about patients who presented appropriate defibrillator shocks during follow-up.

## 5. Conclusions

Compared to patients with non-ischemic etiology, patients with ischemic HF have more comorbidities. Ischemic etiology was not associated with mortality in HF patients. In ischemic HF, the variables associated with a poor prognosis were diabetes in women, and previous readmissions and chronic obstructive pulmonary disease in men. Additional studies are required to determine the specific impact of other etiologies on the prognosis of HF.

## Figures and Tables

**Figure 1 jcm-10-01713-f001:**
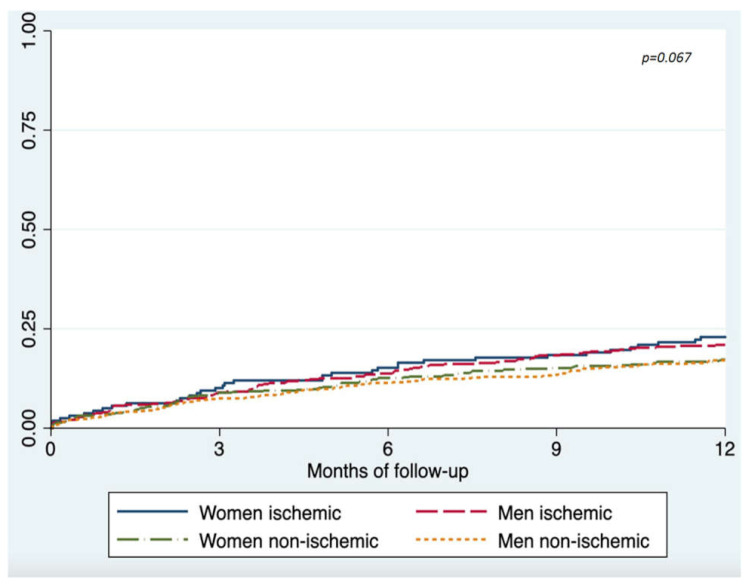
Show Kaplan–Meier survival curves of mortality at 12 months according to HF etiology, in men and women.

**Table 1 jcm-10-01713-t001:** Baseline characteristics in patients admitted with heart failure (HF) according to etiology and sex.

	Ischemic HF (*n* = 613)	Non-Ischemic HF (*n* = 1218)	*p*	Women, Ischemic HF (*n* = 167)	Men, Ischemic HF (*n* = 446)	*p*	Women, Non-Ischemic HF (*n* = 589)	Men, Non-Ischemic HF (*n* = 629)	*p*
Age (years)	73.3 ± 10.6	71.9 ± 12.8	<0.001	77.2 ± 10.3	71.9 ± 10.4	<0.001	74.7 ± 11.7	69.4 ± 13.3	0.002
Previous heart failure diagnosis	415 (67.7)	645 (53.0)	<0.001	121 (72.5)	294 (65.9)	0.283	327 (55.5)	318 (50.6)	0.002
Years since the diagnosis	3.0 ± 5.3	4.0 ± 5.6	0.019	3.4 ± 4.9	4.2 ± 5.9	0.107	3.3 ± 5.8	2.6 ± 4.8	<0.001
Previous heart failure admissions	290 (47.3)	455 (37.3)	0.003	84 (50.3)	206 (46.2)	0.366	220 (37.4)	235 (37.4)	0.997
Tobacco use			<0.001						
- Smoker - Former smoker	74 (12.1)	361 (29.7)		5 (3.0)	69 (15.5)	<0.001	26 (4.4)	307 (49.1)	<0.001
	269 (44.0)	119 (9.8)		17 (10.2)	252 (56.6)	<0.001	54 (9.2)	93 (14.9)	<0.001
Alcohol consumption	54 (8,8)	139 (11.4)	0.002	1 (0.6)	53 (11.9)	<0.001	10 (1.7)	129 (20.5)	<0.001
Dyslipidemia	429 (70.0)	583 (47.9)	<0.001	109 (65.3)	320 (71.8)	0.192	303 (51.4)	280 (44.6)	0.044
Diabetes mellitus	364 (59.4)	483 (39.7)	<0.001	115 (68.9)	249 (55.8)	0.010	244 (41.4)	239 (38.1)	0.463
Hypertension	540 (88.1)	874 (71.8)	<0.001	150 (89.8)	390 (87.4)	0.547	444 (75.4)	430 (68.5)	0.025
Obesity	282 (46.0)	609 (50.0)	0.106	78 (46.7)	204 (45.7)	0.831	299 (50.8)	310 (49.3)	0.606
Chronic kidney disease			<0.001						
- GFR < 30 - GFR 30–59	49 (8.1)	49 (4.1)		15 (9.0)	34 (7.6)	0.332	25 (4.2)	24 (3.8)	0.255
	171 (28.2)	231 (19.1)		42 (25.1)	129 (28.9)	0.220	108 (18.3)	123 (19.6)	0.148
Chronic obstructive pulmonary disease	102 (16.6)	188 (15.5)	0.475	8 (4.8)	94 (21.1)	<0.001	53 (9.0)	135 (21.5)	<0.001
Stroke	67 (10.9)	116 (9.5)	0.646	17 (10.2)	50 (11.2)	0.875	52 (8.8)	64 (10.2)	0.609
Peripheral arterial disease	104 (8.5)	106 (17.3)	<0.001	22 (13.2)	84 (18.8)	0.205	48 (8.2)	56 (8.9)	0.395
Previous myocardial infarction	412 (67.2)	72 (5.9)	<0.001	117 (70.1)	295 (66.1)	0.132	27 (4.6)	45 (7.7)	0.065
Previous coronary artery revascularization	227 (37.2)	55 (4.5)	<0.001						
- Percutaneous - Surgical - Both	83 (13.6)	32 (2.6)		61 (36.5)	166 (37.4)	0.232	24 (4.1)	31 (5.0)	0.929
	54 (8.8)	15 (1.2)	<0.001	20 (12.0)	63 (14.2)	0.210	15 (2.6)	17 (2.7)	0.875
			<0.001	9 (5.4)	45 (10.1)	0.199	7 (1.2)	8 (1.3)	0.935
Atrial fibrillation	201 (32.7)	571 (46.9)	<0.001	56 (33.5)	145 (32.5)	0.811	288 (48.9)	283 (45.0)	0.362
Implantable cardioverter defibrillator	61 (10.0)	66 (5.5)	0.002	4 (2.4)	57 (12.8)	<0.001	16 (2.7)	50 (8.0)	<0.001
Implantable cardioverter defibrillator in patients with a LVEF < 35%	*42* (*15.9*)	*42* (*13.3*)	*0.382*	*2* (*4.8*)	*40* (*95.2*)	*0.003*	*7* (*16.7*)	*35* (*83.3*)	*0.126*
Cardiac resynchronization therapy	24 (3.9)	27 (2.2)	0.130	3 (1.8)	21 (4.7)	0.161	10 (1.7)	17 (2.7)	0.485
Previous treatments:									
ACEIs	290 (47.3)	398 (32.7)	<0.001	77 (46.1)	213 (47.8)	0.517	179 (30.4)	219 (34.9)	0.128
ARBs	153 (24.9)	291 (23.9)	0.373	50 (29.9)	103 (23.1)	0.005	156 (26.5)	135 (21.5)	0.093
Betablockers	430 (70.1)	562 (46.1)	0.001	118 (70.7)	312 (70.1)	0.960	282 (48.0)	281 (44.6)	0.012
Ivabradine	52 (8.5)	35 (2.9)	<0.001	9 (5.4)	43 (9.6)	0.212	13 (2.2)	22 (3.5)	0.358
Loop diuretics	391 (63.8)	687 (56.5)	0.004	104 (62.3)	287 (64.4)	0.639	354 (60.1)	333 (53.0)	0.043
Mineralocorticoid receptor antagonists	182 (29.7)	255 (20.9)	<0.001	38 (22.8)	144 (32.3)	0.004	124 (21.1)	131 (20.8)	0.213
Digoxin	47 (7.7)	151 (12.4)	0.006	12 (7.2)	35 (7.9)	0.894	85 (14.4)	66 (10.5)	0.101
Oral anticoagulation	213 (34.9)	507 (41.7)	0.011	51 (30.5)	162 (36.5)	0.183	275 (46.7)	232 (37.1)	0.001
Aspirin	343 (56.1)	294 (24.2)	<0.001	97 (58.1)	246 (55.4)	0.552	128 (21.7)	166 (26.5)	0.075
Clopidogrel	149 (24.4)	48 (3.9)	<0.001	40 (24.0)	109 (24.6)	0.878	23 (3.9)	25 (4.0)	0.483
Ticagrelor	9 (1.5)	0	<0.001	5 (3.0)	4 (0.9)	0.073	1 (0.2)	0	0.485
Prasugrel	17 (2.8)	1 (0.1)	<0.001	5 (3.0)	12 (2.7)	0.846	0	1 (0.2)	0.250

Data are shown as mean ± standard deviation for continuous variables, and *n* (%) for categorical. ACEI: Angiotensin Converting Enzyme Inhibitors; ARB: Angiotensin II Receptor Blockers; GFR: Glomerular Filtration Rate; LVEF: Left Ventricular Ejection Fraction.

**Table 2 jcm-10-01713-t002:** Vital signs at admission and in-hospital treatments in patients with heart failure (HF) according to etiology and sex.

	Ischemic HF (*n* = 613)	Non-Ischemic HF (*n* = 1218)	*p*	Women, Ischemic HF (*n* = 167)	Men, Ischemic HF (*n* = 446)	*p*	Women, Non-Ischemic HF (*n* = 589)	Men, Non-Ischemic HF (*n* = 629)	*p*
Systolic blood pressure	134 ± 29	135 ± 30	0.8772	140 ± 29	132 ± 30	0.007	136 ± 28	133 ± 29	0.635
Diastolic blood pressure	75 ± 18	76 ± 18	0.167	73.8 ± 17.7	75.5 ± 18.1	0.300	75 ± 17	77 ± 13	0.263
Heart rate	86 ± 22	91 ± 28	0.002	85.1 ± 23.5	85.9 ± 22.1	0.690	91 ± 28	90 ± 27	0.756
Left bundle branch block	97 (16.7)	179 (15.3)	0.006	24 (15.1)	73 (17.3)	0.793	77 (13.7)	102 (16.9)	0.314
QRS duration (ms)	119 ± 33	113 ± 33	<0.006	110 ± 29	123 ± 35	<0.001	108 ± 31	118 ± 34	0.012
Left ventricular diastolic diameter (mm)	58 ± 11	53 ± 11	<0.001	52 ± 9	60 ± 11	<0.001	49 ± 9.6	57 ± 10	0.076
Left ventricular ejection fraction (%) during admission	41 ± 17	48 ± 18	<0.001	47 ± 18	39 ± 16	<0.001	54 ± 16	43 ± 18	0.039
Moderate-severe mitral regurgitation	203 (33.3)	401 (32.9)	0.524	51 (30.5)	152 (34.0)	0.622	189 (32.0)	212 (33.7)	0.393
Sodium (mEq/L)	138 ± 5	139 ± 5	0.236	138 ± 5	139 ± 4	0.070	139 ± 4	139 ± 5	0.051
Potassium (mmol/L)	4.4 ± 0.7	4.3 ± 0.7	0.033	4.4 ± 0.8	4.3 ± 0.7	0.261	4.3 ± 0.7	4.3 ± 0.7	0.077
Glomerular filtration rate (mL/min)	60.2 ± 29.4	67.2 ± 30.3	<0.001	61.8 ± 29.9	55.8 ± 27.4	0.024	64.1 ± 29.4	70.1 ± 31.0	0.188
NT-proBNP	947 ± 1295	747 ± 904	0.001	990 ± 1363	929 ± 1267	0.657	670 ± 858	829 ± 944	0.051
Haemoglobin (g/L)	12.2 ± 20.1	12.3 ± 21.1	0.011	11.4 ± 14.9	12.3 ± 21.2	<0.001	11.8 ± 17.8	12.8 ± 22.7	<0.001
Non-invasive mechanical ventilation	36 (5.9)	62 (5.2)	<0.001	6 (3.7)	30 (6.8)	0.085	25 (4.3)	37 (6.0)	0.226
Invasive mechanical ventilation	6 (1.0)	8 (0.7)	0.075	1 (0.6)	5 (1.1)	0.198	2 (0.3)	6 (1.0)	0.210
Intraaortic balloon pump	3 (0.5)	2 (0.2)	<0.001	1 (0.6)	2 (0.5)	0.276	0	2 (0.3)	0.387
Treatments at hospital discharge:									
ACEIs	339 (55.3)	579 (47.6)	0.006	90 (53.9)	249 (55.8)	0.634	250 (42.4)	329 (52.4)	<0.001
ARBs	101 (16.5)	225 (18.5)	0.324	37 (22.1)	64 (14.4)	0.014	119 (20.2)	106 (16.9)	0.265
Betablockers	479 (80.5)	782 (66.3)	<0.001	127 (77.9)	352 (81.5)	0.354	282 (47.8)	281 (44.7)	0.012
Ivabradine	70 (11.4)	64 (5.3)	<0.001	13 (7.8)	57 (12.8)	0.124	22 (3.7)	42 (6.7)	0.030
Loop diuretics	512 (83.5)	1001 (82.3)	0.792	143 (85.6)	369 (82.7)	0.495	496 (84.2)	505 (80.4)	0.180
Mineralocorticoid receptor antagonists	269 (43.9)	525 (43.1)	<0.001	62 (37.1)	207 (46.4)	0.003	192 (33.9)	181 (30.2)	<0.001
Digoxin	56 (9.1)	254 (20.9)	<0.001	16 (9.6)	40 (9.0)	0.814	134 (22.8)	120 (19.1)	0.287
Oral anticoagulation	266 (44.7)	696 (58.9)	<0.001	66 (40.2)	200 (46.4)	0.176	369 (64.2)	327 (54.1)	<0.001
Aspirin	369 (62.0)	302 (25.6)	<0.001	101 (61.6)	268 (62.2)	0.495	123 (21.4)	179 (29.6)	0.001
Clopidogrel	174 (29.2)	72 (6.1)	<0.001	48 (29.3)	126 (29.2)	0.994	27 (4.7)	45 (7.4)	0.018
Ticagrelor	13 (2.2)	2 (0.2)	<0.001	8 (4.9)	5 (1.2)	0.010	0	2 (0.3)	0.035
Prasugrel	9 (1.5)	1 (0.1)	<0.001	3 (1.8)	6 (1.4)	0.712	0	1 (0.2)	0.069
Percutaneous coronary artery revascularization during admission	66 (11.0)	35 (2.9)	<0.001	17 (10.4)	49 (11.2)	0.651	12 (2.1)	23 (3.7)	0.171
Surgical coronary artery revascularization during admission	6 (0.5)	12 (2.0)	0.013	1 (0.6)	11 (2.5)	0.172	3 (0.5)	3 (0.5)	0.726
Death during hospital admission	45 (3.7)	26 (4.3)	0.851	8 (4.8)	18 (4.0)	0.202	20 (3.4)	25 (4.0)	0.703
Length of hospital stay (days)	11.1 ± 9.5	11.3 ± 16.4	0.837	8.8 ± 5.5	12.0 ± 10.5	0.002	10.1 ± 9.1	12.4 ± 21.0	<0.001

Data are shown as mean ± standard deviation for continuous variables, and *n* (%) for categorical. ACEI: Angiotensin Converting Enzyme Inhibitors; ARB: Angiotensin II Receptor Blockers; NT-proBNP: N-terminal (NT)-pro hormone BNP.

**Table 3 jcm-10-01713-t003:** Events during follow-up in patients admitted with heart failure (HF) according to etiology and sex.

	Women, Ischemic HF (*n* = 167)	Men, Ischemic HF (*n* = 446)	*p*	Women, Non-Ischemic HF (*n* = 589)	Men, Non-Ischemic HF (*n* = 629)	*p*
Hospital readmissions due to HF at 12 months	57 (34.1)	146 (32.7)	0.773	181 (30.7)	159 (25.3)	0.034
All-cause mortality at 12 months	42 (25.2)	102 (22.9)	0.593	111 (18.9)	120 (19.1)	0.918
Heart transplant at 12 months	1 (0.6)	13 (2.9)	0.050	4 (0.7)	18 (2.9)	0.004
Sudden cardiac death at 12 months	4 (2.4)	18 (4.0)	0.331	10 (1.7)	23 (3.7)	0.035
Death due to refractory heart failure	19 (11.4)	52 (11.7)	0.923	59 (10.0)	44 (7.0)	0.058
Death due to non-cardiac causes	13 (7.8)	24 (5.4)	0.259	25 (4.2)	36 (5.7)	0.293

Data are shown as mean ± standard deviation for continuous variables, and *n* (%) for categorical.

**Table 4 jcm-10-01713-t004:** Independent predictors of mortality, and mortality/readmissions at 12 months in patients admitted with heart failure (HF) according to etiology and sex.

Women, Ischemic HF		
**12-Month Mortality**	**HR (95% CI)**	***p***
- ACEIs/ARBs	0.58 (0.24–0.97)	*0.041*
**12-Month Mortality/Readmissions**	**HR (95% CI)**	***p***
- Diabetes mellitus	2.23 (1.05–4.47)	0.035
**Men, Ischemic HF**		
**12-Month Mortality**	**HR (95% CI)**	***p***
- Age	1.03 (1.01–1.05)	0.010
- ACEIs/ARBs	0.58 (0.33–0.79)	0.002
**12-Month Mortality/Readmissions**	**HR (95% CI)**	***p***
- Previous HF admissions	1.15 (1.02–1.29)	0.022
- Chronic obstructive pulmonary disease	1.20 (1.02–1.41)	0.026
**Women, Non-Ischemic HF**		
**12-Month Mortality**	**HR (95% CI)**	***p***
- Age	1.04 (1.14–1.06)	0.002
- Diabetes mellitus	1.34 (1.13–1.58)	0.001
- Chronic kidney disease	1.28 (1.07–1.52)	0.008
- ACEIs/ARBs	0.46 (0.31–0.69)	<0.001
**12-Month Mortality/Readmissions**	**HR (95% CI)**	***p***
- Previous HF admissions	1.25 (1.14–1.38)	<0.001
**Men, Non-Ischemic HF**		
**12-Month Mortality**	**HR (95% CI)**	***p***
- Age	1.05 (1.03–1.67)	<0.001
- Previous HF admissions	1.88 (1.03–1.67)	0.004
- Chronic kidney disease	1.22 (1.08–1.39)	0.002
- ACEIs/ARBs	0.61 (0.40–0.91)	0.015
**12-Month Mortality/Readmissions**	**HR (95% CI)**	***p***
- Age	1.02 (1.01–1.03)	0.021
- Previous HF admissions	1.28 (1.17–1.39)	<0.001
- ACEIs/ARBs	0.69 (0.50–0.97)	0.033

ACEI: Angiotensin Converting Enzyme Inhibitors; ARB: Angiotensin II Receptor Blockers; CI: Confidence Interval; HF: Heart Failure; HR: Hazard Ratio.

## Data Availability

Data will be provided by request.

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
