# Peer review of "Ischemic Etiology and Prognosis in Men and Women with Acute Heart Failure"

_jcm, 2021, doi:10.3390/jcm10081713_

Round 1
Reviewer 1 Report
The authors well responded the reviewer's comment.
Author Response
We would like to thank Reviewer 1 by the comments that have helped us to improve our manuscript.Reviewer 2 Report
In the paper "Ischemic etiology and prognosis in men and women with heart failure" Vicent and coll analyzed the impact of ischemic etiology on prognosis, among men and women hospitalazed for acute HF
The study is well organized and conducted and the paper is well written. Overall the paper represent a very excellence example of gender medicine.
Miinor points
- Since the population study consists of patients with acute heart failure I suggest to change the title in " Ischemic etiology and prognosis in men and women with acute heart failure"
- Please revise Table 3- Please add at table 3 number of patients with implant of left ventricular assist device at 1 month, 6 months and 12 months as well as number of patients with appropriate ICD shocks.
- In order to characterize better the study population please add at table 1 type and dosage of disease modifying drugs
- Please add a section on study limitations
Author Response
In the paper "Ischemic etiology and prognosis in men and women with heart failure" Vicent and coll analyzed the impact of ischemic etiology on prognosis, among men and women hospitalazed for acute HF
The study is well organized and conducted and the paper is well written. Overall the paper represent a very excellence example of gender medicine.
Thank you for this comment.
Minor points
- Since the population study consists of patients with acute heart failure I suggest to change the title in " Ischemic etiology and prognosis in men and women with acute heart failure"
We have changed the title according to the Reviewer suggestion as follows: “Ischemic etiology and prognosis in men and women with acute heart failure”.
- Please revise Table 3- Please add at table 3 number of patients with implant of left ventricular assist device at 1 month, 6 months and 12 months as well as number of patients with appropriate ICD shocks.
We do not have information regarding the number of patients who received a left ventricular assist device implantation, and the rate of appropriate ICD shocks was not recorded.
- In order to characterize better the study population please add at table 1 type and dosage of disease modifying drugs
The dosage of disease modifying drugs was not recorded. Table 1 includes the type of disease modifying drugs as follows:
|
|
Ischemic HF (N= 613) |
Non-ischemic HF (N=1218) |
P |
Women, ischemic HF (N= 167) |
Men, ischemic HF (N=446) |
P |
Women, non-ischemic HF (N=589) |
Men, non-ischemic HF (N= 629) |
P |
|
Previous treatments: ACEIs ARBs Betablockers Ivabradine Loop diuretics Mineralocorticoid receptor antagonists Digoxin Oral anticoagulation Aspirin Clopidogrel Ticagrelor Prasugrel |
290 (47.3) 153 (24.9) 430 (70.1) 52 (8.5) 391 (63.8) 182 (29.7)
47 (7.7) 213 (34.9) 343 (56.1) 149 (24.4) 9 (1.5) 17 (2.8) |
398 (32.7) 291 (23.9) 562 (46.1) 35 (2.9) 687 (56.5) 255 (20.9)
151 (12.4) 507 (41.7) 294 (24.2) 48 (3.9) 0 1 (0.1) |
<0.001 0.373 0.001 <0.001 0.004 <0.001
0.006 0.011 <0.001 <0.001 <0.001 <0.001 |
77 (46.1) 50 (29.9) 118 (70.7) 9 (5.4) 104 (62.3) 38 (22.8)
12 (7.2) 51 (30.5) 97 (58.1) 40 (24.0) 5 (3.0) 5 (3.0) |
213 (47.8) 103 (23.1) 312 (70.1) 43 (9.6) 287 (64.4) 144 (32.3)
35 (7.9) 162 (36.5) 246 (55.4) 109 (24.6) 4 (0.9) 12 (2.7) |
0.517 0.005 0.960 0.212 0.639 0.004
0.894 0.183 0.552 0.878 0.073 0.846 |
179 (30.4) 156 (26.5) 282 (48.0) 13 (2.2) 354 (60.1) 124 (21.1)
85 (14.4) 275 (46.7) 128 (21.7) 23 (3.9) 1 (0.2) 0 |
219 (34.9) 135 (21.5) 281 (44.6) 22 (3.5) 333 (53.0) 131 (20.8)
66 (10.5) 232 (37.1) 166 (26.5) 25 (4.0) 0 1 (0.2) |
0.128 0.093 0.012 0.358 0.043 0.213
0.101 0.001 0.075 0.483 0.485 0.250 |
- Please add a section on study limitations
We have added a section addressing the study limitations: “Our study has some limitations. Follow-up duration was 12 months, and a longer follow-up period may have shown significant differences in outcomes according to HF etiology, and sex. The dosage of disease modifying drugs was not recorded, and we do not have information about patients who presented appropriate defibrillator shocks during follow-up.”